# Suppression of Seedling Survival and Recruitment of the Invasive Tree *Prosopis juliflora* in Saudi Arabia through Its Own Leaf Litter: Greenhouse and Field Assessments

**DOI:** 10.3390/plants12040959

**Published:** 2023-02-20

**Authors:** Ahmed M. Abbas, Maryam M. Alomran, Nada K. Alharbi, Stephen J. Novak

**Affiliations:** 1Department of Biology, College of Science, King Khalid University, Abha 61413, Saudi Arabia; 2Department of Botany and Microbiology, Faculty of Science, South Valley University, Qena 83523, Egypt; 3Department of Biology, College of Science, Princess Nourah bint Abdulrahman University, P.O. Box 84428, Riyadh 11671, Saudi Arabia; 4Department of Biological Sciences, Boise State University, Boise, ID 83725, USA

**Keywords:** days to first emergence, germination percentage, percent viable seeds, allelopathy, bet-hedging strategy, environmental conditions, percent non-viable seeds, population crashes and collapse

## Abstract

Many studies have focused on how leaf litter depth affects seed germination and seedling growth because the seedling stage is the most vulnerable portion of a plant’s life cycle. Invasive plants with the most severe ecological consequences are those that modify ecosystems, and this can occur through the formation of thick litter layers which can suppress the emergence, survival, and recruitment of native plant seedlings; in addition, in some cases, these litter layers can suppress invasive plant seedling recruitment. *Prosopis juliflora* is a thorny shrub that is native to arid and semi-arid portions of North America, parts of South America, and the Caribbean. It has invaded millions of hectares around the world, including Saudi Arabia. The objective of this study is to evaluate whether *P. juliflora* leaf litter reduces the recruitment of its own seedlings under greenhouse and field conditions in Saudi Arabia. In both the greenhouse and the field, the number of days to first emergence increased and germination percentage decreased with increasing litter depth. With the 1, 2, and 4 cm litter depth treatments, the number of viable seeds generally decreased, with no emergence, germination, or viable seeds detected for the 8 cm litter depth treatment. Results of this study reveal that increasing the depth of *P. juliflora* leaf litter suppresses the survival and recruitment of its own seedlings. Future search should assess the actual mechanisms through which *P. juliflora* seeds are suppressed, the role of allelopathic compounds in this process, and whether viable seeds are dormant and will persist in the soil seed bank.

## 1. Introduction

Leaf litter plays an important role in the demographic and ecological dynamics of plant populations and communities [1]. Studies have shown that leaf litter is critical in structuring plant communities because it influences the abiotic and biotic environments that individual plants experience and has both direct and indirect effects [2,3]. Most studies have focused on how leaf litter, especially litter depth, affects seed germination and seedling growth because the seedling stage is the most vulnerable portion of a plant’s life cycle [4]. The seedlings of plants are sensitive to light quantity and quality, temperature (especially extreme fluctuations in temperature), and soil moisture levels, all of which are influenced by the amount of leaf litter [2,5,6]. A sufficient litter layer will maintain soil moisture and reduce the intensity of desiccation, thereby increasing seed germination, seedling survival, and seedling establishment [7]. Thus, litter layers directly influence the demography of plant populations and the structure and dynamics of plant communities [8].

The influence of leaf litter on seed germination, seedling emergence, seedling survival, and seedling recruitment varies with habitat type (environmental conditions), the species producing the litter, the amount of litter (especially litter depth), seed position, and seed traits [5,9,10]. For instance, a meta-analysis conducted by Xiong and Nilsson [11] indicated that leaf litter had a negative effect on seed germination and seedling recruitment in grassland, old field, and forest habitats. Other studies show that low-to-moderate amounts of leaf litter have a positive effect on seedling emergence and recruitment in arid ecosystems, dry grasslands, and during droughts [12,13,14,15]. Moderate amounts of litter can increase seedling survival, growth, and recruitment by modulating microclimate conditions and attenuating extremes in soil moisture and temperature [7,14,16,17,18,19], thus facilitating plant performance and population stability [20].

Conversely, these facilitative effects are reduced, or cease altogether, when the amount of litter is too high [11,21]. Thick litter layers reduce light quality and quantity beneath the litter [22], decrease soil temperature fluctuations [8], create a barrier that does not allow seeds to come into contact with the soil, and/or prevent the roots of seeds that germinate in the litter from reaching the soil [2,8,11,13,15,18,23]. The amount of litter can alter the outcome of plant–plant interactions (i.e., competition), which may lead to changes in the composition and structure of plant communities [8]. Finally, some plants produce exudates or leachates which act as phytotoxins (allelopathic compounds) that can inhibit the growth of other plant species, and if their concentration increases sufficiently, these same compounds can even inhibit the growth of seedlings of the same species (i.e., autotoxicity occurs). The decomposition of leaf litter can release such allelopathic compounds into the environment [24].

Biological invasions occur when individuals are introduced into a new region in which their descendants persist, proliferate, and spread beyond their original points of introduction [25,26]. Many invasive species have caused severe negative ecological consequences in their new range [27], have high economic costs [28], and are a leading threat to biodiversity worldwide [29]. In extreme cases, invasive species can result in the extinction of native species [30,31]. Invasive species degrade ecosystems by altering community structure [32], ecosystem processes [33], nutrient fluxes [34,35], disturbance regimes (e.g., the fire regime) [36,37,38], and hydrological cycles [39]. Due to their negative ecological consequences and high economic costs, invasive species are the focus of extensive research ([25,40]. Much of this research is aimed at understanding (1) which non-native species will become invasive, (2) which native communities will be invaded, and (3) the long-term fate of invasive species [25,40,41,42,43,44,45,46]. 

Invasive plants with the most severe ecological consequences are those that modify ecosystems and cause positive feedbacks that enhance their dominance within native plant communities over time [47,48]. This can occur when invasive plants form extensive monocultural stands, produce large amounts of biomass, and generate an abundant and thick litter layer [36,37,49]. Thick litter layers produced by numerous invasive plant species have been reported to suppress the emergence, survival, and recruitment of native plant seedlings [47,48,49,50,51,52,53,54]. In some cases, the litter of invasive species also contains allelopathic compounds that are released into the environment as their litter decomposes and these compounds inhibit the seedlings of native plant species [51,52,53]. The role of allelopathic compounds in the invasion process is referred to as the “Novel Weapons Hypothesis”, as described by Hierro and Callaway [55] and Hierro et al. [56]. 

The invasion of *Prosopis juliflora* (Sw.) DC. (common names mesquite or velvet mesquite; Fabaceae) in rangeland, semi-arid, and arid habitats around much of the world has occurred because of repeated deliberate introduction events, its ability to tolerate drought and salt stress, the size of its canopy, extensive seed production, seed dispersal through endozoochory, and the production of large amounts of leaf litter which contains allelopathic chemical compounds [53,57,58,59,60,61]. *Prosopis juliflora* is a thorny shrub (1–3 m) or tree (12 m) that is native to arid and semi-arid portions of North America (southwest United States and northwest Mexico), parts of South America, and the Caribbean. It has invaded millions of hectares across northeastern Brazil [62], South and East Africa [60,63,64], the Arabian Peninsula [57,65], India [66], coastal Asia [67], Australia and some Pacific islands [68]. Several *Prosopis* species have been introduced widely and cultivated in many areas around the globe because they have multiple uses (e.g., shade, forage, timber, and food with edible fruits (indehiscent legumes) that can be ground into flour; their individuals (individual plants) also produce multiple stems which allow several harvests of the foliage each year [59,69]). *Prosopis* is now considered one of the world’s worst woody invasive plant taxa [70]. 

In a recent survey of the invasive plant species in Saudi Arabia, Thomas et al. [65] found that *P. juliflora* was one of the six most impactful species; it occurred in all areas that were surveyed, and it was the dominant invasive species of low elevation sites in the country. In the southwestern region of the Arabian Peninsula, *P. juliflora* has invaded the edge of farms, stony deserts, open plains, sand sheets, rivers, flood plains, and wadies [71]. In such habitats, the water table is close to the soil surface. In southwestern Saudi Arabia, *P. juliflora* is reported to decrease native plant biodiversity and alter ecosystem processes [65,71]. In addition to natural areas, *P. juliflora* was also reported as a ruderal species of urban and suburban areas in Saudi Arabia, where it was widely planted as a shade tree and now forms dense thickets [65].

While the invasion of *P. juliflora* in Saudi Arabia and elsewhere around the world occurs, at least in part, due to the large amounts of litter it produced and the allelopathic chemicals it contains, there are also reports that the emergence, survival, and recruitment of *P. julifora* seedlings is inhibited through its own litter [53,72]. The overall objective of this study is to evaluate whether *P. juliflora* leaf litter reduces the recruitment of its own seedlings. This was accomplished by adding various depths of leaf litter to *P. juliflora* seeds and measuring seed germination and viability parameters under greenhouse and field conditions. We predict that the germination and viability of *P. juliflora* seeds will be reduced with the increasing depth of its own litter.

## 2. Results

### 2.1. Prosopis Juliflora Seed Characteristics

The mean volume of the *P. juliflora* seeds used in these experiments was 56.7 ± 0.63 mm^3^ (Table 1). The mean seed mass of the seeds was 0.041 ± 0.001 g (range = 0.033–0.048 g).

### 2.2. Greenhouse and Field Leaf Litter Experiment Conditions

In the greenhouse, litter load significantly increased with litter depth (*r* = −0.980, *p* < 0.001, *n* = 20) (Table 2). Soil pH increased significantly with leaf litter depth, from 6.25 ± 0.10 for the control treatment (no litter) to 7.83 ± 0.09 for the 8 cm litter depth treatment (*r* = 0.96, *p* < 0.01, *n* = 20), and EC decreased significantly from 0.44 ± 0.01 for the control treatment to 0.17 ± 0.01 mS·cm^−1^ for the 8 cm litter depth treatment (*r* = −0.98, *p* < 0.01, *n* = 20; Table 2). The daily range of soil temperatures (maximum–minimum temperatures) decreased significantly with increasing litter depth (Table 2), indicating the insulative effects of increasing leaf litter depth. Maximum daily soil temperatures in the greenhouse decreased significantly with litter depth and ranged from 35.4 ± 0.4 °C for the control treatment to 24.3 ± 0.3 °C for the 8 cm litter depth treatment (*r* = −0.956, *p* > 0.001, *n* = 20). Minimum daily temperature decreased with litter depth (*r* = −0.94, *p* < 0.001, *n* = 20; Table 2), except for litter depth 0 cm and 1 cm and litter depth 4 cm and 8 cm. 

In the field, the maximum daily soil temperature ranged from 46.8 ± 0.5 °C for the control treatment to 22.6 ± 0.7 °C for the 8 cm litter depth treatment, and these values decreased significantly with increasing litter depth (*r* = −0.992, *p* < 0.001, *n* = 20; Table 2). Minimum daily soil temperatures decreased significantly with increasing litter depth (*r* = −0.98, *p* < 0.0001, *n* = 20). The daily range of soil temperatures (maximum–minimum temperatures) decreased significantly with the increased litter depth (*r* = −0.99, *p* < 0.001, *n* = 20; Table 2).

### 2.3. Seedling Emergence, Seed Germination, and Seed Viability Measurements

The number of days to first emergence differed between the greenhouse and field experiments, varying from 4 days for the control treatment (no litter) to 18 ± 1.9 days for the 4 cm litter depth treatment in the greenhouse (Figure 1); and varying from 2 ± 0.2 days for the treatment control to 12 ± 1.8 days for the 4 cm litter depth treatment in the field (Figure 2). The number of days to first emergence increased significantly with litter depth in both the greenhouse and field experiments (greenhouse: *F* = 12.86, *p* < 0.01, *n* = 12; field: *F* = 26.05, *p* < 0.0001, *n* = 12). 

Germination percentage (germination %) of *P. juliflora* in the greenhouse and the field for the control treatment (no litter) was approximately 60% and 20%, respectively (Figure 1 and Figure 2), whereas these germination percentages increased to 85 ± 5% and 38 ± 3%, respectively, for the 1 cm litter depth treatment. Germination percentage in greenhouse and field decreased significantly with increasing litter depth (greenhouse: *F* = 45.69, *p* < 0.001; field: *F* = 10.50, *p* < 0.001) (Figure 1 and Figure 2). No *P. juliflora* seeds germinated at a litter depth of 8 cm either in the greenhouse or in the field. With litter cover, germination percentages in the greenhouse and field increased as the daily maximum temperature increased (greenhouse: *r* = 0.83, *p* < 0.0001, *n* = 16; field: *r* = 0.27, *p* < 0.0001, *n* = 16) (Table 2).

The largest values for percentage of viable seeds, 38% and 80%, occurred for the control treatments in the greenhouse and field experiments, respectively (Figure 1 and Figure 2). The percentage of viable seeds decreased with 1 cm of litter depth, increased with 2 cm of litter depth, and decreased again with 4 cm of litter depth. The decrease in percentage of viable seeds for the 1 cm litter depth treatment occurred, at least in part, because the germination percentage was highest for this litter depth. In the greenhouse and the field, the percentage of non-viable *P. juliflora* seeds increased with increasing litter depth, with 100% non-viable seeds recorded for the 8 cm litter depth treatment (Figure 1 and Figure 2). The percentage of viable seeds decreased with increasing maximum daily soil temperature in the greenhouse experiment (*r* = −0.24, *p* < 0.05, *n* = 16); however, in the field experiment, the percentage of viable seeds increased with increasing maximum daily soil temperature (*r* = 0.93, *p* < 0.001, *n* = 16).

In the greenhouse experiment, the percentage of non-viable seeds for the control treatment and the 1 cm litter depth treatment were both 0.0%, indicating that all seeds under these two conditions had either germinated or were still viable (Figure 1). However, the number of non-viable seeds was statistically significant and increased steadily for the 2, 4, and 8 cm litter depth treatments, with the 8 cm litter depth treatment having 100% non-viable seeds. In the field experiment, only the control treatment had a non-viable seed percentage of 0.0%, and the number of non-viable seeds showed a statistically significant increase for all litter depth treatments (Figure 2). Similarly, the 8 cm litter depth treatment had 100% non-viable seeds.

## 3. Discussion

Non-native woody plant species (shrubs and trees) often produce an abundant and thick leaf litter layer which can inhibit the germination, survival, and recruitment of native seedlings in native plant communities [47,48,49,50,51,52,53,54]. For many of these non-native species, the suppressive effects of a thick litter layer are amplified by the production and release of allelopathic compounds from this accumulated litter layer and other tissues (i.e., the “Novel Weapons Hypothesis”) [51,52,53,55,56]. While the production of abundant amounts of litter and allelopathic chemicals contributes to invasiveness, there is growing recognition that these traits can also reduce the germination and survival of invasive plant seedlings and lead to recruitment failures in populations of invasive plants, as reported for *Alliaria petiolata* (M.Bieb.) Cavara and Grande (garlic mustard) in Ohio, USA [50] and *Rhododendron ponticum* L. (common rhododendron) in Ireland [51]. Similar findings have been reported for *P. juliflora* in Ethiopia [53] and Saudi Arabia [72]. The results of the current study are consistent with these earlier reports; the litter of *P. juliflora*, especially deeper litter depths, reduces the emergence of its own seedlings, and reduces seed germination and viability. Our findings suggest that, under some circumstances, invasive populations of *P. juliflora* may crash or collapse due to the self-regulatory effects associated with an abundant and thick leaf litter layer. 

The effects of leaf litter on plant populations and communities are determined by a complex set of interacting factors: habitat type, which species are present, litter depth or abundance, the chemical composition of litter, seed characteristics, and the position of seeds in the litter [9,10,11,24,47,48,49,50,51,52,53,54]. Chief among these factors is litter abundance and depth [47,48,49,50,51,52,53,54]. Thus, in this study, we tested the effects of four *P. juliflora* leaf litter depths (1–8 cm) on seedling emergence, seed germination, and seed viability of *P. juliflora*, under greenhouse and field conditions. 

In both the greenhouse and field experiments, a significantly smaller number of days to emergence of the first seedlings was observed for the control treatments, i.e., in the absence of any leaf litter (Figure 3 and Figure 4). The number of days to first emergence steadily increased with litter depth (1, 2, and 4 cm), with no seedlings emerging from the 8 cm litter depth treatment. In addition, fewer days to first emergence were observed under field conditions compared with greenhouse conditions, suggesting that some aspect of the field environment promotes greater seedling growth rates and thus fewer days to emergence. Our results also indicate that with 1, 2, and 4 cm of litter depth, seedlings of *P. juliflora* were able to elongate their shoots to a sufficient height to emerge above the litter layer (this did not occur with 8 cm of litter depth). Finally, an increased number of days to first emergence means that longer periods of time are required for seedlings to emerge from the litter and gain access to full sunlight. These longer periods of time are likely to negatively impact seedling survival and recruitment. 

We detected slightly different patterns for germination percentage in the greenhouse and field: (1) germination percentages were generally higher in the greenhouse experiment than in the field experiment; (2) in both the greenhouse and the field, germination percentage values were significantly larger for the 1 cm litter depth treatments compared to the control (no litter) treatments; (3) germination percentages decreased with increasing litter depth (2 and 4 cm); and (4) no seed germination occurred with the 8 cm litter depth treatment (Figure 3 and Figure 4). Our results therefore suggest that leaf litter depth has a two-fold effect on germination percentages in *P. juliflora*. First, the presence of a relatively small amount of litter (1 cm) appears to promote seed germination. Second, increasing litter depths reduces seed germination and large amounts of litter (8 cm litter depth) can eliminate germination completely. Similar results have been reported in studies assessing the influence of leaf litter on seed germination of many native plant species [47,48,49,50,51,52,53,54]. 

For the control (no litter) and 1 cm litter depth treatments in the greenhouse, and the control treatment in the field, the values of germination percentage and percentage of viable seeds summed to 100%, and no seeds were determined to be non-viable (i.e., no seeds died) (Figure 1 and Figure 2). This was nearly true for the 1 cm litter depth treatment in the field experiment (<10% of seeds were non-viable). These results suggest that the environment conditions associated with these two treatments, in both the greenhouse and the field, were optimal, or close to optimal, and allowed seeds to either germinate or remain viable (e.g., they may have entered dormancy). The sum of germination percentage and percentage of viable seeds decreased more sharply in the field compared to the greenhouse, suggesting that leaf litter depth was interacting with some unknown factor(s) to reduce seed survival and germination more rapidly under field conditions. However, whether in the greenhouse or the field, the conditions associated with the 8 cm litter depth treatments did not allow any seeds to germinate or survive (percentage of non-viable seeds = 100%). The results of this study clearly indicate that future research should investigate the exact mechanism(s) through which an increase in the depth of *P. juliflora* litter reduces the germination, emergence, and survival of its own seeds and seedlings. 

Litter depth (i.e., litter load) not only influences seed germination and seedling growth and recruitment by forming a physical barrier [9,10,11], but litter can also affect the chemical properties of soil and it can alter microclimate conditions and extremes in soil temperature and moisture [14,16,17,18,19,20]. Under greenhouse conditions, we found that soil pH increased with litter depth (Table 1), which may be due to increased Mg^2+^ and Ca^2+^ input into the soil and may improve the buffering capacity of soil, as reported by Finzi et al. [73]. Electrical conductivity (EC) is a measure of the soils’ ability to conduct electricity, and is related to the total concentration, mobility, valence, and relative proportion of ions in the soil sample. Values of EC in the greenhouse ranged from 0.44 to 0.17 mS·cm^−1^, with these values decreasing with increasing litter depth (Table 1). Because *P. juliflora* is capable of tolerating and may require high salt concentrations [74], reductions in seed germination and seedling growth and recruitment reported here may be influenced by the reduced soil ion (salinity) concentrations associated with increasing litter depth. Seed germination in *P. juliflora* is influenced by a complex set of factors, including interactions among different levels of salinity, temperature, and light [75]. For instance, at lower salinity levels, seed germination rates were higher at 40 °C than at 15 and 25 °C. Based on the soil temperature results reported here (Table 1), it is difficult to interpret the role of temperature on *P. juliflora* seed germination and seedling growth recruitment. However, the highest seed germination percentages were for the control (no litter) and the 1 cm litter depth treatments in the greenhouse and in the field, for which soil temperature values were highest. 

The presence of a litter layer can also maintain higher soil moisture levels and reduce seed and seedling desiccation [14,16,17,18,19,20,76], which facilitates seed germination and seedling establishment in an arid environment, or during drought [12,13,14,15]. Leaf litter depth can also influence both the quality (wavelength) and quantity of light that seeds receive under the litter layer [77], and light levels have been shown to play a role in the germination of *P. juliflora* seeds [76]. Unfortunately, we did not measure moisture and light levels in this study; however, the sum of the environmental conditions described above must have been optimal for seed germination under the 1 cm litter depth treatment, which would explain why the greatest germination percentage values were observed for this litter depth. Similarly, most of the litter depths deployed in this study created environmental conditions that were sufficient to maintain the viability of seeds that did not germinate. Such seeds may be dormant, and, under field conditions, are likely to incorporated into the soil seed bank. 

## 4. Materials and Methods

### 4.1. Fruit and Leaf Litter Collection

*Prosopis juliflora fruits* were collected in March 2021 from multiple mature individuals chosen at random from Wadi Qaradah in Southwest Saudi Arabia (18°14′25″ N–42°51′17″ E; 1090 m; Figure 3 and Figure 4). Monospecific stands of *P. juliflora* in Wadi Qaradah measured approximately 10–12 ha. The DBH of the trees in this study area ranged between 21 and 36 cm and tree height ranged between 8 and 10 m. After harvest, fruits were stored at 5 °C, under dry conditions and in the dark, until these experiments were conducted. Leaf litter was also obtained from Wadi Qaradah by collecting dead stems and leaves of several *P. juliflora* plants. The leaf litter was checked for the presence of *P. juliflora* seeds and fungal pathogens. None was found. We attempted to ensure that the seed–leaf litter interactions examined in this study were at an ecological equilibrium by collecting *P. juliflora* fruits and litter from the same location.

### 4.2. Greenhouse Leaf Litter Experiment

A greenhouse experiment was conducted from April 2021 to August 2021 (Figure 4). Seeds of *P. juliflora* were removed from pericarps, weighed, and measured (length, width, and thickness) with an electronic Vernier calliper (Quality Control Company, CA, USA; 0.01 mm accuracy) (*n* = 50) to determine average seed mass and volume. Seed volume was calculated as length × width × thickness. 

Four replicates of 25 seeds were sown at 1 cm depth in sand in plastic containers measuring 18 cm width, 22 cm length, and 11 cm height (containing ~1.6 kg of washed sand). A procedural control treatment was set up, which did not include any seeds, to test whether the sand we used contained *P. juliflora* seeds. Five leaf litter depth treatments were conducted: control (bare sand = no litter), 1 cm litter depth (1093 ± 37 g dry weight DW litter m^−2^), 2 cm litter depth (2503 ± 20 g DW litter m^−2^), 4 cm litter depth (3230 ± 92 g DW litter m^−2^), and 8 cm litter depth (8875 ± 68 g DW litter m^−2^, see Table 2). These litter depth treatments were chosen based on field observations at Wadi Qaradah in Southwest Saudi Arabia where *P. juliflora* is very abundant (Ahmed Abbas, personal observation). Observations at Wadi Qaradah were consistent with results reported by Slate et al. [78], who found that, on average, the litter layer beneath dense canopies of *P. juliflora* in the United Arab Emirates had a depth of just over 3 cm, and approximately 2.5 cm at the edge of the canopies. In addition, we observed that the litter layer under large monospecific stands of *P. juliflora* was distributed over a relatively broad area, as reported by Basavaraja et al. [79]. 

This experimental design consisted of five litter depth treatments with four containers (replicates) per depth (five litter depth treatments x four replicates per treatment). Containers were watered gently (to minimize the disturbance of seeds) once a day to ensure that the moisture of the soil remained within 70% of its water-holding capacity. Fresh water (<0.5 Practical Salinity Unit) was used to avoid salinity effects on germination because we wanted to record seed responses to litter depth only. The containers had small holes at the bottom to enable drainage, and these holes were covered with strips of cloth to prevent the loss of sand.

### 4.3. Seedling Emergence, Seed Germination, and Seed Viability Measurements

The number of days to first emergence for each replicate is the time required for the first seedling to emerge from either the soil surface (for the control treatments) or the litter layer (for the leaf litter treatments) [60]. The number of germinating seeds was monitored throughout the duration of the greenhouse and field experiments. At the end of the experiments, leaf litter was carefully removed and seedlings that died before removing the litter were counted. Germination percentage was calculated as the number of seeds that germinated divided by the total number of seeds per replicate. Ungerminated seeds were collected for additional analysis. They were soaked in water at 30 °C for 24 h, an incision was made in their seed coats, and the embryo was soaked in 1% tetrazolium chloride (Panreac Quimica S.A., Barcelona, Spain) for 24 h at 30 °C. Pink-colored embryos were scored as alive, and these seeds were classified as being viable (they are probably dormant). Seeds that did not take up the stain were classified as being non-viable (they may be dead). The percentages of viable and non-viable seeds were calculated in relation to the total number of seeds per treatment [58,59,60]. For each litter depth treatment, the germinated seeds percentage, viable seeds percentage, and percentage of non-viable seeds sum to 100%.

### 4.4. Greenhouse Leaf Litter Environmental Conditions

The pH and electrical conductivity (EC) of the sand potting soil were recorded at the end of the greenhouse experiment, in August 2021. Measurements were carried out at 1 cm depth of the sand. pH was recorded in the laboratory after adding distilled water to the soil (1:1, soil: distilled water, *v*/*v*) (pH/redox Crison with the electrode M-506). Soil salinity was measured as EC (conductivity meter, Crison-522) after pH (1:2, soil: distilled water, *v*/*v*). Daily maximum and minimum soil temperatures, and the range of soil temperatures (maximum–minimum temperatures) were recorded for the 0, 1, 2, 4 and 8 cm litter depth treatments using a digital thermometer (Cole-Parmer GmbH, Wertheim am Main, Germany). Temperature measurements were taken every four hours (from 8 a.m. to 8 p.m.) for a 3-day period, at the end of these experiments. The average daily air temperature during the greenhouse experiment was 23.4 ± 0.5 °C, with a mean low temperature of 12.3 ± 1.6 °C and a mean maximum temperature of 44 ± 3.5 °C. Mean daily relative humidity of the air was 71.5 ± 2.6%, ranging between 43.5 ± 5.5% and 91.5 ± 1.1%.

### 4.5. Field Leaf Litter Experiment

The effects of *P. juliflora* leaf litter on seedling emergence, seed germination, and seed viability were also assessed under field conditions, at Wadi Qaradah in Southwest Saudi Arabia (Figure 4). For this experiment, fruits of *P. juliflora* were collected in April 2021 from multiple mature individuals chosen at random from Wadi Qaradah, and stored at 5 °C, under dry conditions, in the dark, until the experiment was conducted from April 2021 to August 2021. Leaf litter was also obtained from Wadi Qaradah by collecting dead culms and leaves of several *P. juliflora* plants. The leaf litter was checked for the presence of *P. juliflora* seeds and fungal pathogens, and none was found. The leaf litter was then thoroughly mixed, and air-dried for 72 h prior to use. Two 0.40 m^2^ plots per leaf litter treatment were in Wadi Qaradah, under gaps in the *P. juliflora* canopy. Five leaf litter depth treatments were conducted: control (bare ground = no litter), 1 cm litter depth (1000 ± 35 g DW litter m^−2^), 2 cm litter depth (2500 ± 15 g DW litter m^−2^), 4 cm (3500 ± 84 g DW litter m^−2^), and 8 cm litter depth (9000 ± 50 g DW m^−2^). The number of days to first emergence, germination percentage, percentage of viable seeds, and percentage of non-viable seeds were calculated as described above [58,59,60].

### 4.6. Statistical Analysis

All statistical analyses were carried out using SPSS 12.0 (SPSS Inc, Chicago, IL, USA). The data in this study were tested for homogeneity of variance using the Brown–Forsythe test and for normality using the Kolmogorov–Smirnov test, respectively (*p* < 0.05). Statistically significant differences in the number of days to first emergence, germination percentage, percentage of viable seeds, and percentage of non-viable seeds for the litter depth treatments were tested using one-way analysis of variance (ANOVA, *F*-test). The Tukey honest significant difference (HSD) test was used to determine significant differences among litter depth treatment means only if the *F*-test was significant (*p* < 0.05) [59,60,61]. The associations between leaf litter load, pH, EC, and soil temperature values for the litter depth treatments were assessed using the Pearson correlation coefficient and linear regression analyses.

## 5. Conclusions

The results of this study reveal that increasing depth of *P. juliflora* leaf litter suppressed the survival and recruitment of its own seedlings. To better understand this phenomena, future research should assess the actual mechanisms through which *P. juliflora* seeds and seedlings were suppressed. For instance, are seeds and seedlings killed by the barrier created by the litter, or by the alteration of environmental conditions under the litter, or both mechanisms? In addition, the role of allelopathic compounds contained in the litter of *P. juliflora* should be explicitly examined. Allelopathy, along with the other two mechanisms, likely all interact in combination to suppress the seeds and seedlings of *P. juliflora*. Finally, many viable seeds were observed for the 1, 2 and 4 cm litter depth treatments, and the status of these viable seeds should be determined by future research. For instance, are these seeds dormant? If they are, will they enter the soil seed bank? Storage of *P. juliflora* seeds in the soil seed bank could serve as a bet-hedging strategy that allows them to await better conditions for germination.

## Figures and Tables

**Figure 1 plants-12-00959-f001:**
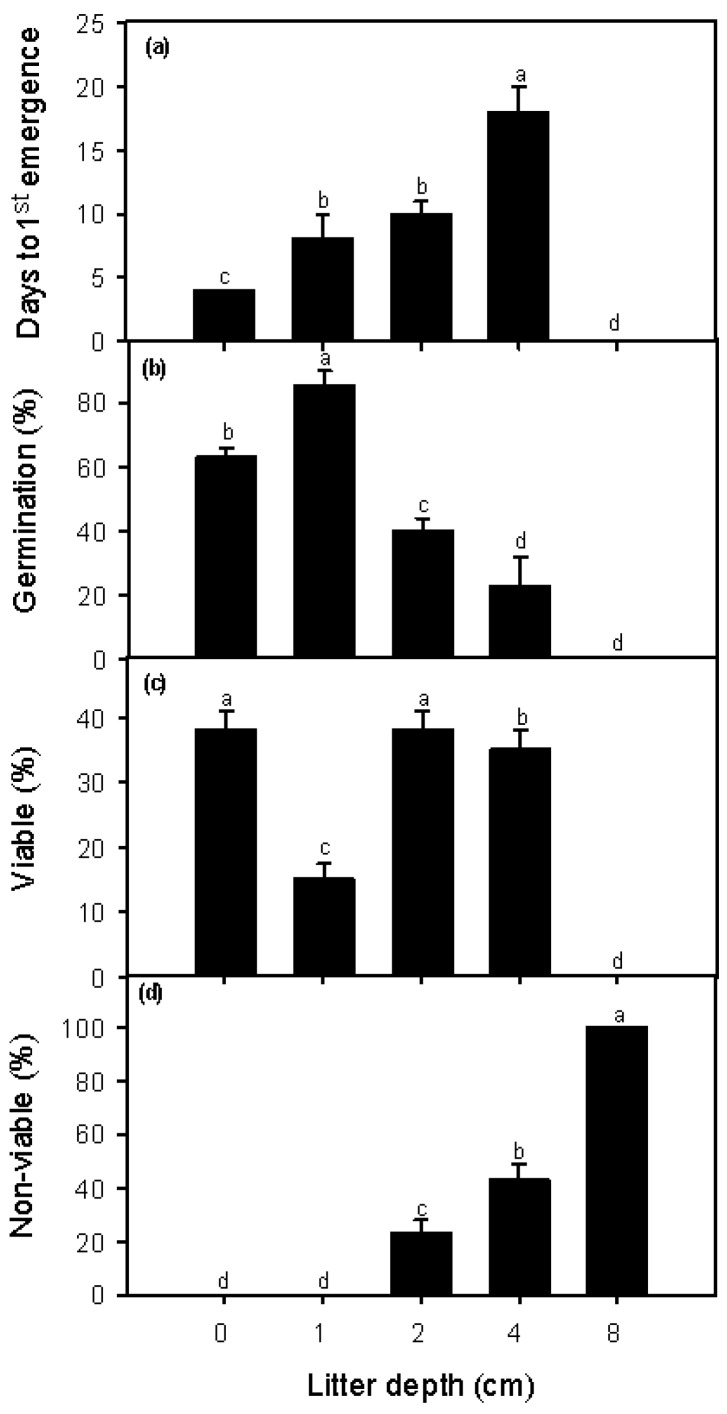
Days to first germination (**a**), germination percentage (**b**), and percentages of viable (**c**) and non-viable (**d**) seeds of *Prosopis julflora* for the five leaf litter depth treatments under greenhouse conditions. The value of germination percentage, present viable seeds, and percentage of non-viable seeds sums to 100% for each litter depth. Different letters indicate significant differences between treatments (ANOVA, *p* < 0.05).

**Figure 2 plants-12-00959-f002:**
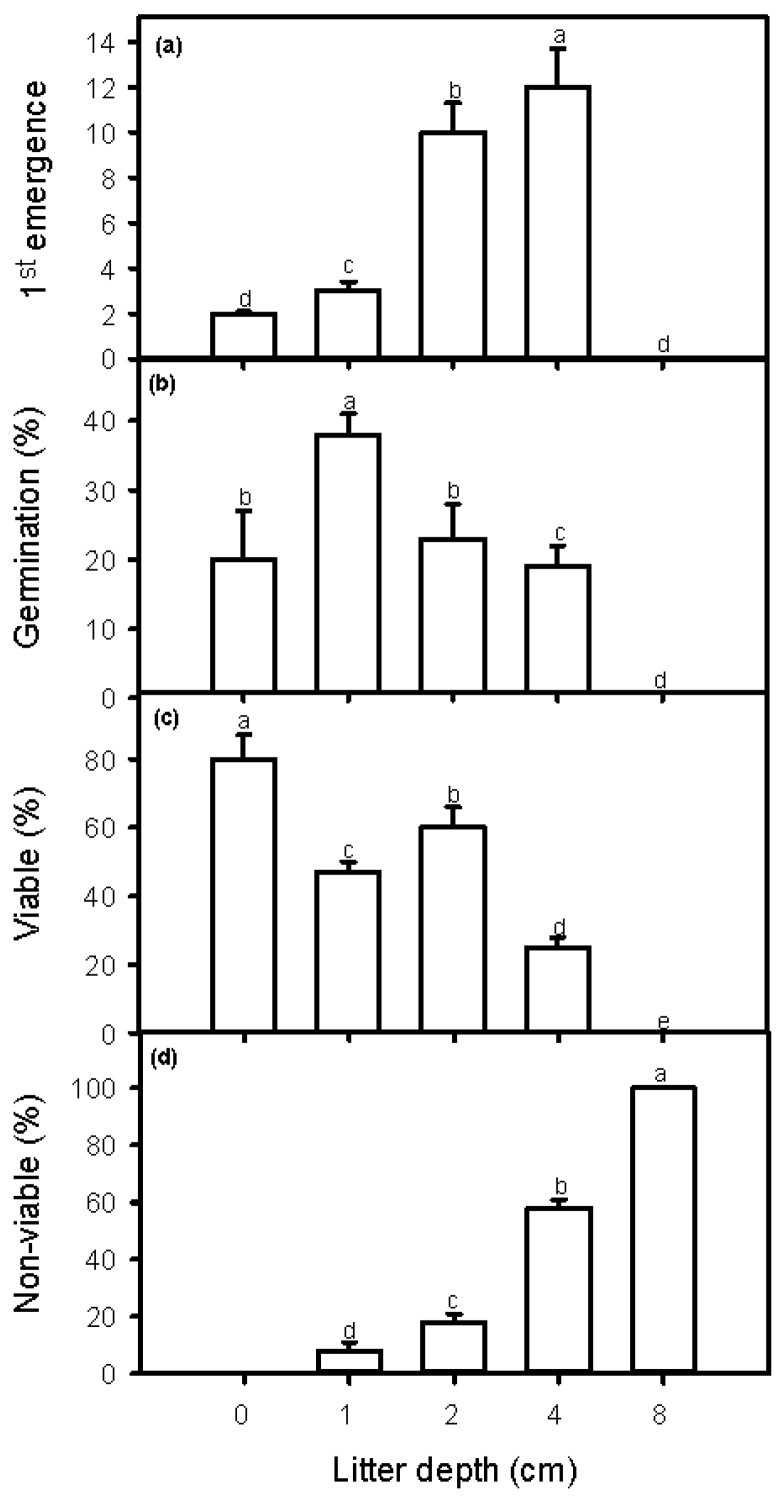
Days to first germination (**a**), germination percentage (**b**), and percentages of viable (**c**) and non-viable (**d**) seeds of *Prosopis julflora* for the five leaf litter depth treatments under field conditions. The value of germination percentage, present viable seeds, and percentage of non-viable seeds sums to 100% for each litter depth. Different letters indicate significant differences between treatments (ANOVA, *p* < 0.05).

**Figure 3 plants-12-00959-f003:**
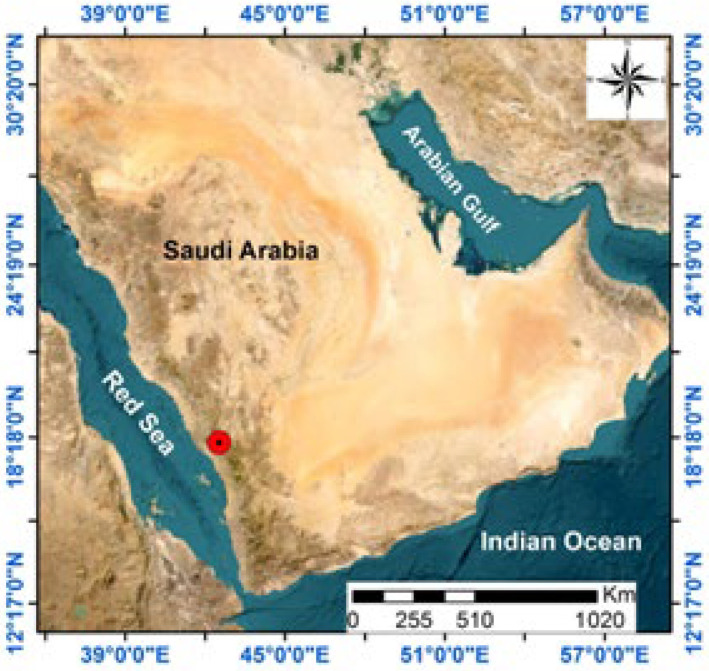
Map showing the location of Wadi Qaradah in Southwest Saudi Arabia, indicated with the red pinpoint.

**Figure 4 plants-12-00959-f004:**
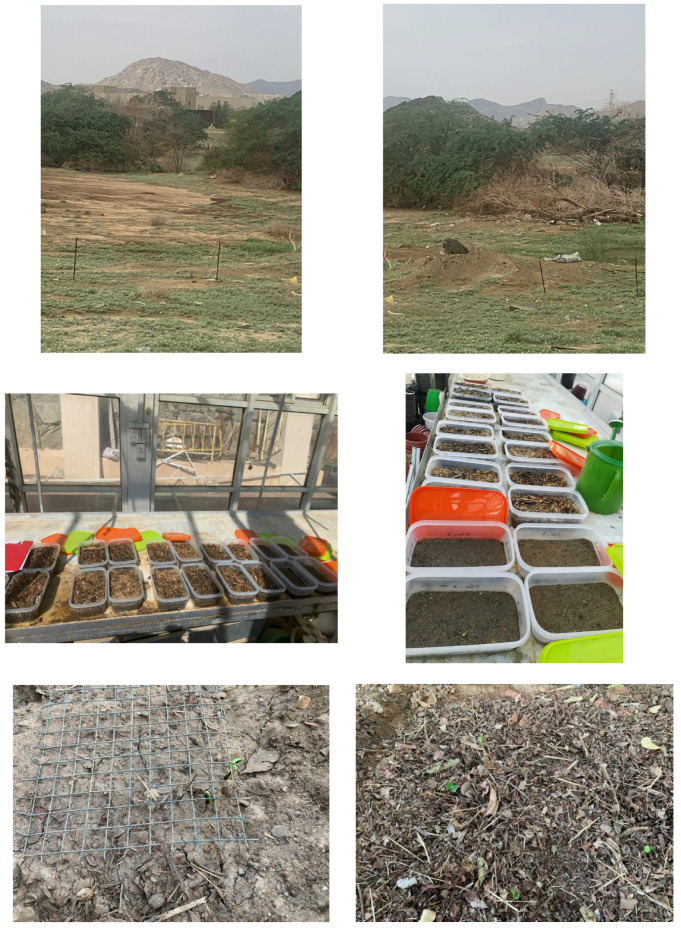
Photographs of *Prosopis juliflora* and the greenhouse and field experiments conducted in this study. The top two photos show *Prosopis juliflora* from Wadi Qaradah, Southwest Saudi Arabia. The two photos in the middle of this figure show the set-up of the greenhouse experiment (see the text for a further explanation). The bottom two photos show the setup of the field experiment (**left**) and the field plots with a litter layer and *Prosopis juliflora* seedlings that have emerged from the litter (**right**).

**Table 1 plants-12-00959-t001:** Seed morphological characteristics (mm) and seed mass (mg) for the invasive tree *Prosopis juliflora* from Wadi Qaradah, Southwest Saudi Arabia. Values are mean ± SE (*n* = 50 seeds).

Length (mm)	Width (mm)	Height (mm)	Volume (mm^3^)	Mass (g)
6.3 ± 0.05	3.9 ± 0.03	2.2 ± 0.02	56.7 ± 0.63	0.041 ± 0.001

**Table 2 plants-12-00959-t002:** Litter load (g m^−2^), soil pH, electrical conductivity (mS·cm^−1^), maximum and minimum soil temperature (°C), and range of soil temperatures (maximum–minimum temperatures) (°C) (*n* = 3) for the five *Prosopis juliflora* leaf litter depth treatments. Different letters indicate significant differences between treatments (ANOVA, *p* < 0.05).

Litter Depth (cm)	Litter Load (g·m^−2^)	pH	Electrical Conductivity (mS·cm^−1^)	Daily Soil Temperature (°C)					
				Green House			Field		
				Maximum	Minimum	Max–Min	Maximum	Minimum	Max–Min
0	0 ± 0 ^e^	6.25 ± 0.10 ^e^	0.44 ± 0.01 ^a^	35.4 ± 0.4	21.5 ± 4.3 ^a^	13.9 ± 4.6 ^a^	46.8 ± 0.5 ^a^	24.3 ± 2.1 ^a^	22.5 ± 0.5 ^a^
1	1093 ± 37 ^d^	6.58 ± 0.13 ^d^	0.41 ± 0.01 ^b^	33.5 ± 0.3 ^b^	20.4 ± 1.8 ^a^	13.1 ± 3.4 ^a^	37.5 ± 0.5 ^b^	22.8 ± 0.4 ^a^	14.7 ± 0.5 ^b^
2	2503 ± 20 ^c^	6.83 ± 0.05 ^c^	0.34 ± 0.01 ^c^	30.5 ± 0.2 ^c^	18.0 ± 1.1 ^b^	12.5 ± 2.7 ^a^	35.6 ± 0.2 ^c^	21.3 ± 0.5 ^a,b^	14.4 ± 0.7 ^b^
4	3230 ± 92 ^b^	7.30 ± 0.13 ^b^	0.29 ± 0.02 ^d^	27.7 ± 0.7 ^d^	17.5 ± 1.4 ^c^	10.2 ± 2.8 ^b^	29.5 ± 0.6 ^d^	18.0 ± 1.4 ^b^	11.5 ± 1.9 ^c^
8	8875 ± 68 ^a^	7.83 ± 0.09 ^a^	0.17 ± 0.01 ^e^	24.3 ± 0.3 ^e^	17.3 ± 1.5 ^c^	7.0 ± 1.9 ^c^	22.6 ± 0.7 ^e^	14.5 ± 1.4 ^c^	8.1 ± 2.1 ^d^

## Data Availability

Not applicable.

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
