# Peer review of "Suppression of Seedling Survival and Recruitment of the Invasive Tree Prosopis juliflora in Saudi Arabia through Its Own Leaf Litter: Greenhouse and Field Assessments"

_plants, 2023, doi:10.3390/plants12040959_

Round 1
Reviewer 1 Report
The experimental data are scarce, and it is suggested to add molecular experiments to fill the paper and support the conclusion, and finally to analyze the mechanism.
Reviewer 2 Report
In their manuscripts “Suppression of Seedling Survival and Recruitment of the Invasive Tree Prosopis juliflora in Saudi Arabia by its Own Leaf Litter: Greenhouse and Field Assessments” the authors explore the results of sound experments examining a key demographic process in an important invasive species. Prosopis juliflora, as the authors point out, has been included in lists of the most impactful woody invasive plants both globally and in their study region, Saudi Arabia. Like many dryland invasive species, P. juliflora, modifies its ecosystem when it establishes by increasing the depth and composition of leaf litter. Given reported allelopathic effects, the authors examine whether the species leaf litter suppresses the germination of its own seeds using paired greenhouse and field germination experiments. By tracking the fates of seeds in replicate plots, where they also examined some chemical and microclimate features, they show that increasing leaf litter depth tends to increase pH, decrease CEC and temperature ranges. Seedlings showed the highest germination rate under a thin layer of litter, and no germination under the thickest layer of litter. The authors interpret the germination suppression with the deepest litter as evidence that this species limits its own population growth rate by eliminating recruitment in established stands. They speculate that its populations may actually decline, as has been demonstrated for other invasive plant species with allelopathic properties.
Generally, I found the system compelling, the design strong, the evidence convincing and the interpretation consistent. That said, I believe the authors could improve the text in a few key areas.
Major comments:
First and foremost, the authors may exercise more caution in speculating about a demographic collapse in this species. Individuals probably have a long lifespan and may simply replace one another in closed stands. Importantly, the authors state that the litter depth treatment levels (i.e. upto 8 cm) were based on field surveys. The associated citation differs from others and the abstract available for that article does not report any results for litter depth surveys. Consequently, it is difficult to determine whether the range of experimental conditions are representative. Some more detailed justification, included how frequently these depths are encountered in the field, would be helpful for contextualizing the biological relevance of the results.
Second, the authors did collect a strong multivariate dataset. Some aspects of their analysis are reported inconsistently, including missing statistics and uncertainties in parts of the results (see more detailed comments below). To extract even more information from their dataset, the authors may consider a more powerful multivariate analysis, structural equation modelling (https://www.usgs.gov/centers/wetland-and-aquatic-research-center/science/quantitative-analysis-using-structural-equation), which can quantify the effects of instrumental environmental variables, like soil chemistry and temperature, on the germination rates.
Minor comments:
Abstract: The abstract is concise and informative.
Introduction: The introduction effectively contextualizes the research question and study system. Overall, it was clearly written. However, it could be more concise. Lines 71-95 are informative but read more like a review of invasion biology rather than a framework for this specific study.
Line 39: “have”->”has”; the subject appears to be “leaf litter” / “it”
Line 126: “will be” -> “was”.
Results:
Figures 1 and 2: The description of the response variables does not clearly indicated that germinated viable and non-viable are exhaustive and mutually exclusive outcomes that sum to 100% as explained in the methods. The authors should consider clarifying by expanding the caption to include this information or modifying the y axis titles to “germinated”, “viable, non-germinated” and “unviable, non-germinated”. Also, it would be helpful to explain in the caption at what point the germination percentage was calculated. Otherwise, it is difficult to infer the difference between percent germination and percent viable.
Lines 169-174: Please report uncertainty with the mean estimates for days to emergence. In the parenthetical for the statistics, please report the name of the statistical test, the value of the test statistic (i.e. r, regression beta, t or F) along with the p-values.
Discussion: Like the introduction, the Discussion is well written but may be longer than necessary. The first section goes into great detail on the basic results and could be streamlined.
Line 243: “canopy depth” is a fragment.
Line 253: The parenthetical is redundant.
Methods
Line 340: “culm” generally refers to stems of graminoids, not trees. Perhaps “stems”?
Line 374: This important reference is cited differently from others. Please standardize.
Line 379: Please spell out the abbreviation “psu”.
Line 393: Please clarify that days to first emergence was measured per replicate, not per treatment or for the experiment overall.
Line 402-404: Interpretations for both staining outcomes include “dormant”. Given the usual way of interpreting this test, I believe that the authors can remove dormant from the “unstained” condition.
Lines 446-448: It is unclear how Pearson correlations and linear regressions test differences between these quantitative variables. Aren’t they tests for association?
Reviewer 3 Report
I was presented with a very professional article for review. The text is very interesting and informative at the same time. It brings a solid knowledge of the forest ecology. All elements of this article are properly defined. The best part of this manuscript is the results. In my opinion, this article can be published after some corrections.
Major comments:
- First, the authors need to move the Materials and methods section, from section 4 to section 2.
- Discussion section: Overall, the discussion part is too long. The Discussion should summarize the manuscript's main finding(s) in the context of the broader scientific literature and address any study limitations or results that conflict with other published work. The methodology limitation should be mentioned in the last paragraph of this section.
Minor comments:
L 77-79: Authors need to mention, invasion species also change hydrological cycles, like rainfall partitioning (https://doi.org/10.1016/j.agrformet.2017.03.017), and transpiration (https://doi.org/10.1016/j.pce.2018.10.002).
L 336: Add some information about the characteristic of of seed trees, e.g., DBH, Tree height.
L 338: Add elevation above sea level of the study area.
Author Response
Please see he attached file.

Round 2
